# communications
# engineering

# Spatiotemporal management of solar, wind and hydropower across continental Europe

Anders Wörman [1✉], Ilias Pechlivanidis [2], Daniela Mewes [1], Joakim Riml [1] & Cintia Bertacchi Uvo[3,4]

Weather climate fluctuations cause large variations in renewable electricity production, which requires substantial amounts of energy storage to overcome energy drought periods. Based on daily hydroclimatic data and information about renewable power systems covering Europe, here we quantify the complementarity in the solar-wind-hydro energy components of the continental climate system. We show that the spatiotemporal management of renewable electricity production over Europe can induce a virtual energy storage gain that is several times larger than the available energy storage capacity in hydropower reservoirs. The potential electricity production matches the consumption by spatiotemporal management of suitable shares of solar and wind power complemented with the present hydropower. While the mixed renewable energy potential varies less than anticipated at the continental scale, utilization of the complementarity requires new continental electrical transmission lines and stable international trade. We highlight that management models need to consider incentives beyond national boundaries to appropriately benefit from continental climate conditions.

[1] KTH – Royal Institute of Technology, S-100 44 Stockholm, Sweden. [2] Swedish Meteorological and Hydrological Institute, S-601 76 Norrköping, Sweden. [3] Finish Environmental Institute, FI-00790 Helsinki, Finland. [4] Lund University, S-221 00 Lund, Sweden. ✉email: worman@kth.se

Climate fluctuations reflected in atmospheric and oceanic teleconnections can cause significant variation in the access to renewable energies, from daily erratic behavior to periodic patterns with durations of many years to decades[1–4]. Both short-term intermittency and long-lasting low renewable energy availability can potentially cause energy droughts[5–7], which may lead to electric supply failure with a great impact on society[8]. A key problem in the effective and efficient transition to a renewable energy production system is related to the provision of sufficient energy storage[9,10], especially to overcome long-term climate fluctuations and periods of energy droughts[5]. This problem was well observed in Europe in 2022, during which the severe drought affected the continent's economic and energy crises. Ultrahigh-voltage supergrids transmitting electricity over distances of 2000–3000 km over and between continents with very low losses[11] offer new paths to reduce the energy storage demand by continental balancing of the energy system and to secure the supply of electricity to consumers from renewable energy sources. Continental or intercontinental management of the energy system also requires a shift in the current national concern for energy droughts[8] to a concern about technological prerequisites and management methods that are appropriate at the continental scale or even larger management areas[12,13].

Current models for the engineering design of multienergy production systems generally ignore many details of operational trade-off strategies, and several studies have pointed to essential gaps in the literature in terms of considering the role of distributed generation across continents[11,14,15]. Although operational models normally consider the trade-off between different reasons for economical profit and cost components, including both production revenue, valuation of the stored energy and economical penalties[16–19], there are still several shortcomings in such management optimizations, i.e., (a) the common economic controls in the energy market are divided into short- and long-term markets, and operational planning generally suboptimizes electricity production by using objective functions that are bound to a geographical region, a specific energy resource and/or a particular power company's collection of power plants[20]; (b) long-term decisions in the market are based on climatic predictions, which have limited ability to represent all covariances of solar, wind and hydropower on the spectrum of relevant frequencies[21]; and (c) there are insufficient energy market incentives for considering common system benefits, such as virtual storage services[22]. Some emerging principles have been presented for a market of virtual power trading[23], including the concept of virtual energy storage resulting from balancing consumption with production[24–26]. However, there are no assessments of the potential for virtual energy storage that arises from the spatiotemporal coordination of renewable energy production at the continental scale, which could indicate the value of this resource for different shares and distributions of power sources. In turn, such information is important for developing virtual energy storage market rules and adapting future designs of renewable electricity production systems.

In particular, there is a need to couple the design of the power plant distribution, energy source mix and transmission lines with the operational models of the system-specific benefits. Planning the distribution of power system networks can enhance power system security and facilitate production flexibility[27]. The introduction of increasingly more wind and solar power leads to increasing problems with curtailments, but market-based dispatch mechanisms can reduce those curtailments[20]. Similarly, it has recently been argued that seasonal hydroclimatic predictions can increase the effectiveness of production management[6]. Recent findings show that both wind and solar intensity exhibit reverse correlations with different teleconnection indices in northern and southern Europe, suggesting that their variances are significantly lower when aggregated on continental scales compared to their regional behavior[4]. This finding suggests that virtual energy storage arises primarily from management over large regions, hence accounting for the auto-covariation and the cross-covariations of the energy components inherent in the climatic system. Hence, the key is to apply management models accounting for spatiotemporal coordination and complementarity of renewable energy mixes in operational procedures[28] and for system design at subcontinental or continental scales[29].

Schlachtberger et al.[30] demonstrated for the existing electrical grid topology and a single data year how increasing transmission restrictions increased the energy storage requirement in the balancing of solar, wind and hydropower over Europe. Tong et al.[31] used 39 years of daily data of solar and wind data for 42 countries over the globe to show how the aggregation of solar and wind power significantly increases the reliability to meet electricity demand with different energy storage levels. Here, we analyze the statistical bounds of virtual energy storage for various mixes of solar, wind and hydropower production without transmission limitations and show quantitatively how spatiotemporal coordination generates virtual energy storage that continuously increases with the scale of aggregation as well as the selected regulatory time horizon. We are using several decadal-long historical records of hydroclimatic indicators available at high spatiotemporal resolution covering Europe and parts of the Middle East (Fig. 1a). The study uses spectral decomposition to demonstrate the potential role of energy source complementarity over Europe arising at daily to multi-annual frequencies of importance to balance the energy system. This approach moves beyond a "silo energy source" investigation, as it has been the focus till date, and instead addresses the spatiotemporal sharing and complementarity of renewable energy mixes. The use of historical records of hydroclimatic data from the Copernicus ECMWF database represents the observed covariance patterns between energy components and the spatiotemporal pattern, which could otherwise present a problem in weather fluctuation predictions. We show that suitable shares of solar PV, wind and hydropower combined with spatiotemporal coordination of production across Europe can induce virtual energy storage gain (VSEG) that widely exceeds that available in the current hydropower reservoirs. The entire potential electricity production can precisely match the consumption by spatiotemporal management and using the present hydropower in complement with suitable shares of solar and wind power. Previous studies on virtual energy storage have considered multienergy systems on a river basin size of 128,000 square kilometers[32] or single energy sources[33]. However, here, we reveal the significant enhancement of the national potentials for complementarity between renewable energy mixes induced as the spatiotemporal coordination is extended across Europe over an area of 13,600,000 km$^2$, hence emphasizing the VESG that may arise in the international energy trade. These findings present a striking conceptual advance in our understanding of the potential for trading renewable power across Europe. Our recommendations aim to achieve an integrated, secure and sustainable energy supply at the core of the EU's energy policy, which promotes the utilization of the national potential for renewable energy while simultaneously facilitating cross-border support schemes (EU directive 2018/2001, article 22).

## Results

### Virtual energy storage gain for PV solar, wind and hydropower over Europe

Renewable energy production potentials aggregated over Europe show high short-term intermittency and seasonal variations, which is reflected in the coefficient of variation in the

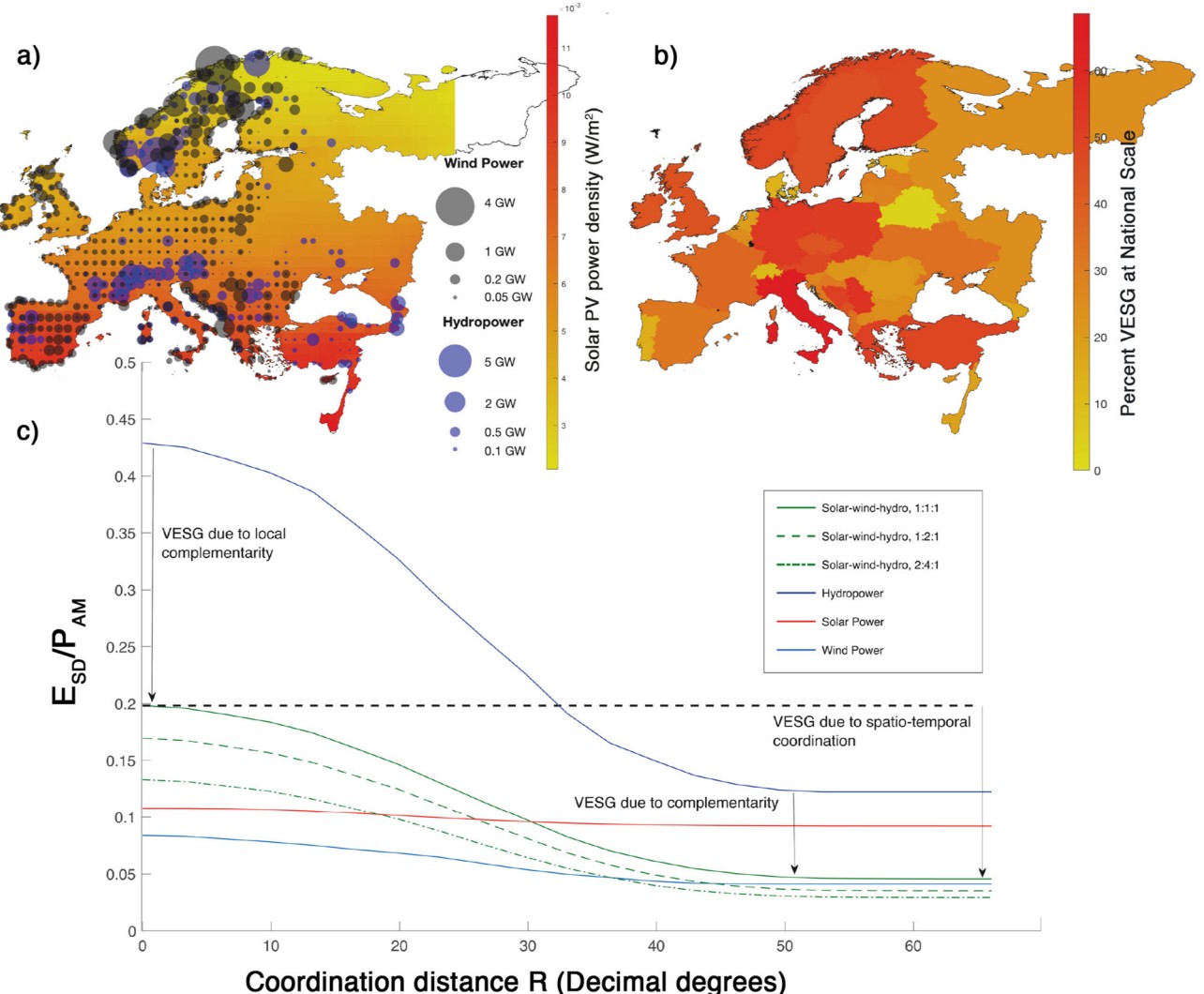

**Fig. 1 Scenarios for power production and energy storage. a** A technical scenario with equal annual production potential for wind, solar and hydropower; the latter is kept constant at the current hydropower production capacity of 642 TWh y$^{-1}$ (blue circles). Scenarios with increasing shares of wind power at the current wind farm locations (gray circles) and an increasing share of uniformly distributed solar PV power (yellow-to-red map background) are used to emphasize the degree of complementarity between renewables. **b** Virtual energy storage gain (VESG) for the scenario of 1:1:1 shares of solar, wind and hydropower due to spatiotemporal coordination at the intranational level is 39% on average in comparison to the energy storage demand without coordination. The corresponding VESG due to international spatiotemporal coordination is 77%. **c** The lower diagram shows how increasing the coordination distance R for renewable energy production across Europe can induce substantial virtual energy storage gain (VESG), which adds to the complementarity gain between renewables. Both complementarity and spatiotemporal coordination contribute significantly to reducing the problem associated with climate-driven energy storage demand in renewable energy production; hence, they facilitate the transition towards renewable energy production.

individual potentials for PV solar power, wind power and hydropower: 0.58, 0.27 and 0.36, respectively. The power spectrum of the solar power potential is lower overall than that of the hydropower and wind power potentials except at the annual peaks that appear for all energy sources (Fig. 2a); this finding suggests the overall lowest variance in solar power (except at the annual peak). All three power spectra exhibit magnitude variations over all frequencies, suggesting that continental access to renewable energy is characterized by periodicities in addition to a significant noise (randomness) at the highest frequencies. When all power sources are aggregated in 1:1:1 shares, the coefficient of variation decreases to 0.22, which is below that of the individual resources, suggesting possible complementarity effects in the European energy system. While this result is interesting, it defines neither the cumulative amount of energy deficits nor the energy storage demand, which is addressed by spectral decomposition.

The complementarity can first be explained by the covariance between the renewable energy sources and second by the covariance between the production potential and the energy storage demand (Methods section "Spectral decomposition of the power potential and the energy storage demand"). At a landscape point or at a regional scale, the potential for PV solar power shows a generally negative covariation with both wind power and hydropower; however, this behavior is modified when the resource variability is assessed over a large region, i.e., Europe. In particular, the North Atlantic Oscillation induces an oscillating inverse behavior of dry-cold and wet-warm weather conditions in the northern and southern parts of Europe[4]. Here, we found that the cross-covariance spectra of wind power versus solar power show a strong negative correlation in the annual period for the spatial distribution of power production (Fig. 2b). Similarly, both wind and hydropower have a positive covariance with the

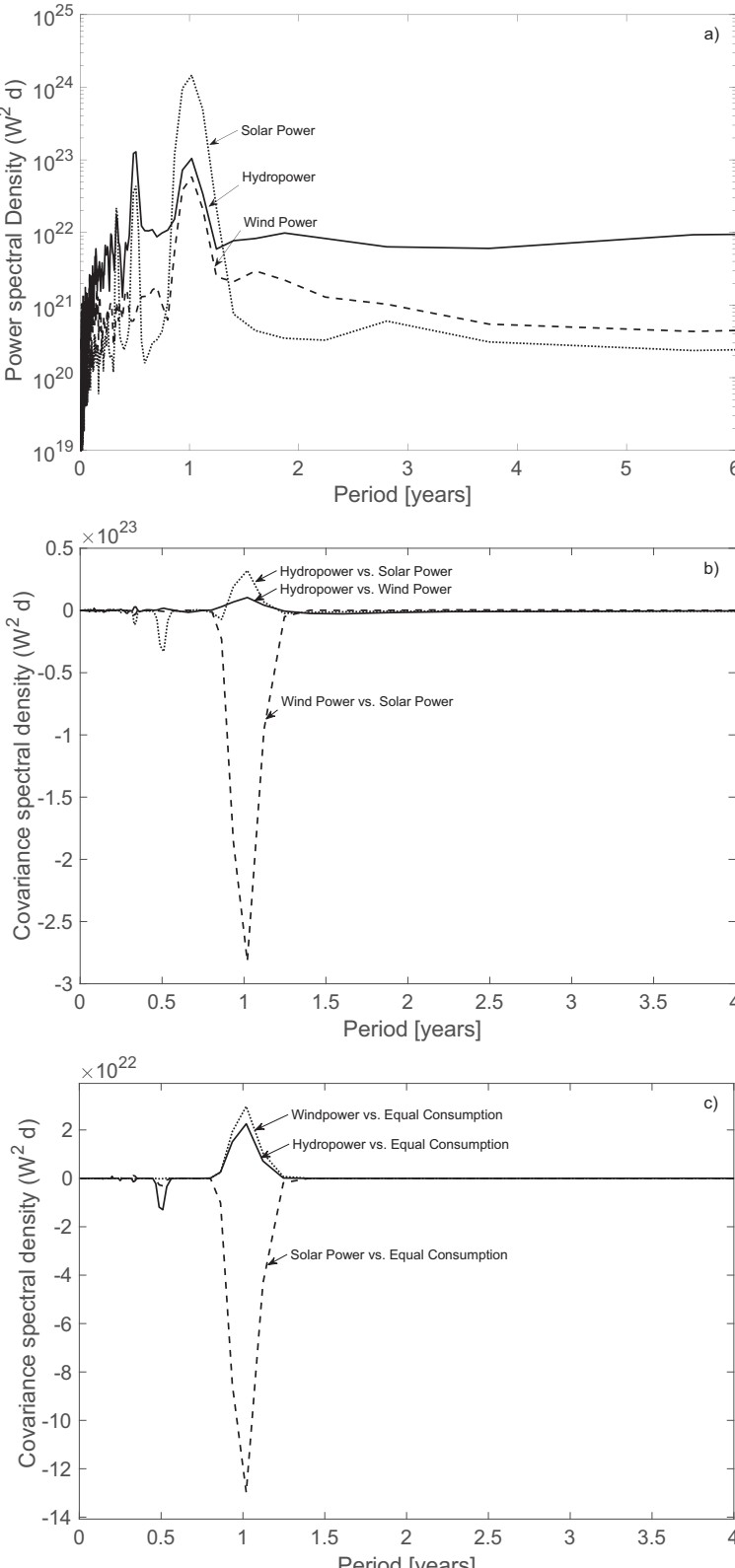

**Fig. 2 Cross- and autocovariance spectra for solar, wind and hydropower. a** Power spectral density of the potential for solar, wind and hydropower production averaged over Europe for the scenario of equal production capacity of each energy source corresponding to 642 TWh y$^{-1}$. **b** Covariances between the energy sources averaged over Europe. **c** Covariance spectra between the potential for different renewables and equal mean consumption following the pattern identified in the EU19 countries from Eurostat data.

electricity demand of Europe (Fig. 2c), which is positive from a management standpoint. In contrast, solar power demand shows a negative correlation with electricity consumption demand. The strong annual periodicity of the covariances among all three resources indicates a strong complementarity for periods of a year or longer.

Moreover, a positive cross-covariance was found between the EU19 consumption pattern and the potential for hydropower and wind power (Fig. 2c). Due to the complementarity among solar, wind and hydropower (expressed by their cross-covariances), there are striking implications for the energy storage demand when these renewables are managed through spatiotemporal coordination. We show that the energy storage demand ($E_{SD}$) decreases significantly across the European continent with increasing coordination distance (see diagram in Fig. 1c). Here, the energy storage demand is normalized with the annual power output, $P_{AM}$ (J or Wh), for each of the demonstrated scenarios. The reduction in the energy storage demand, here referred to as the virtual energy storage gain (VESG), arises due to both (a) the spatiotemporal coordination of production and (b) the complementarity of different shares of solar, wind and hydropower. The VESG can be seen as a measure of a statistical bound of the storage gain if spatiotemporal management and complementarity is fully utilized within the coordination distance.

We find that the current hydropower system producing 642 TWh $y^{-1}$ has an energy storage demand of 275 TWh when used as a solitary resource without complementarity and without spatiotemporal coordination. However, this can be reduced to 67 TWh $y^{-1}$ by spatiotemporal coordination over the analyzed domain. We also note that the complementarity within the solar-wind-hydropower mixture increases the VESG potential.

For the scenario of 1:1:1 shares of solar, wind and hydropower (solid green curve in Fig. 1c), the VESG corresponds to 294 TWh, thus far exceeding that of the energy storage capacity of hydropower reservoirs in Europe, which has been estimated to be approximately 183 TWh. By testing various shares of wind and solar power as complements to the existing hydropower system, we find that the share of 2:4:1 is close to a maximized VESG of 467 TWh, and these shares satisfy the energy production-consumption of 4494 TWh $y^{-1}$. The VESG arising from spatiotemporal coordination of the individual renewable sources corresponds to about 298 TWh, so one could say that the complementarity contributes (467–298=) 169 TWh to the total VESG 467 TWh, whereof 298 TWh is due to spatiotemporal coordination. The energy storage demand is 131 TWh at full spatiotemporal coordination and is thus sufficiently low to be covered by the existing storage capacity of hydropower reservoirs with large margins. Supplementary Note 4 in SI illustrates the contribution to energy storage of hydropower storage, spatiotemporal management and complementarity (183:298:169) for the solar-wind-hydro 2:4:1 scenario. In this scenario, it is clear that all three matching mechanisms are needed to fulfill electricity consumption at all times and locations. In this part our results contrast those of[12] who found that a renewable energy system has lower cost on the continental scale than at regional or national scale, but based on the assumption that energy storage is provided using hydropower storage and a significant amount of hydrogen storage in overground steel tanks. Here the focus is on statistically identifying the VESG without considering the limitations of optimization routines and the exact objectives guiding the routine to maximize it.

**Transition strategy for the future PV solar power, wind power and hydropower production system.** It has been argued that the renewable energy transition can contribute positively to international security and peace but can also exacerbate security due to critical curtailment factors[34]. The geopolitical consequences of changing energy markets are yet to be seen[35]; however, it is clear that market-based dispatch can offer tools for reducing curtailment levels, as has been seen in practice in existing energy markets[20]. In contrast, a lack of energy storage can significantly enhance curtailment levels in renewable energy systems, and therefore, a proportional growth of energy storage with an increasing share of renewables plays an important role in the effective transition and management of renewable electrical systems[18].

Here, we assumed energy markets recognizing spatiotemporal coordination at only the national level (Fig. 1b) and found that the sum of VESG arising within nations in the covered domain was 154 TWh, which is approximately half of the VESG arising from spatiotemporal coordination within the entire domain (294 TWh). In economic terms, the installation cost for an alternative physical device providing the same amount of energy storage as the gain of continental spatiotemporal coordination, i.e., (294–154=) 140 TWh, corresponds to approximately US\$ 280 trillion (using US\$ 2000 KWh$^{-1}$ from Schmidt et al.[18]) or 10 times the yearly GNP of the USA. The relative VESG, percent-VESG = $100 \times$ VESG/$E_{SD}(R = 0)$, which can be induced at national levels, varies from only a few percent up to nearly 60% with an average of 39%. A similar gain of 41.7% was found for the multienergy system of the Yalong River basin in China, which is similar in size as Greece (the UOS-M scenario presented by Jing et al.[32]). These numbers can be compared to the percent-VESG of 77% obtained across the European continent for the scenario of 1:1:1 shares of solar, wind and hydropower.

It is complex to assess how well the current market rules promote a transition towards renewable energy system-supporting services, such as VESG, especially considering the fragmented energy market with multiple price interactions[36] and the significant curtailment currently being observed in renewable energy production when integrated with fossil and nuclear production[20]. A future renewable energy system will likely include more renewable energy sources, such as geothermal and renewable biofuel, as well as nuclear, which will affect the covariances in aggregated energy production; hence, the potential for VESG within solar-wind-hydropower is affected. However, by focusing on the energy sources that are mostly affected by climate fluctuations, our study provides an estimate of the bounding need for energy storage associated with those renewables.

Figure 3 demonstrates the $E_{SD}$ of different expansion scenarios for solar, wind and hydropower and marks the current production in solar, wind and hydropower as well as the current energy storage capacity in hydropower reservoirs. We find that an optimal share of renewables—leading to the least possible energy storage demand—is approximately 3:1 for wind and solar power, but the difference between 2:1 and 3:1 is minor and depends on the exact spatial distributions of power plants across Europe. The optimal share of renewables depends on the general negative correlation between solar and wind power and on the weak correlation that both sources show with hydropower potential (Fig. 2b). Another important factor is how well the distribution of power plants is adapted to the cross- and autocovariance patterns of renewables and to the cross-covariances between production potential and demand (Fig. 2c).

Previous analyses showed that spatiotemporal coordination of hydropower production alone induces a VESG on the continental scale, which is twice that of the existing hydropower reservoirs[33]. The complementary effects of solar, wind and hydropower enhance the VESG even further, which in principle makes it possible to keep the energy storage demand well below the storage capacity of existing hydropower reservoirs (section "Virtual

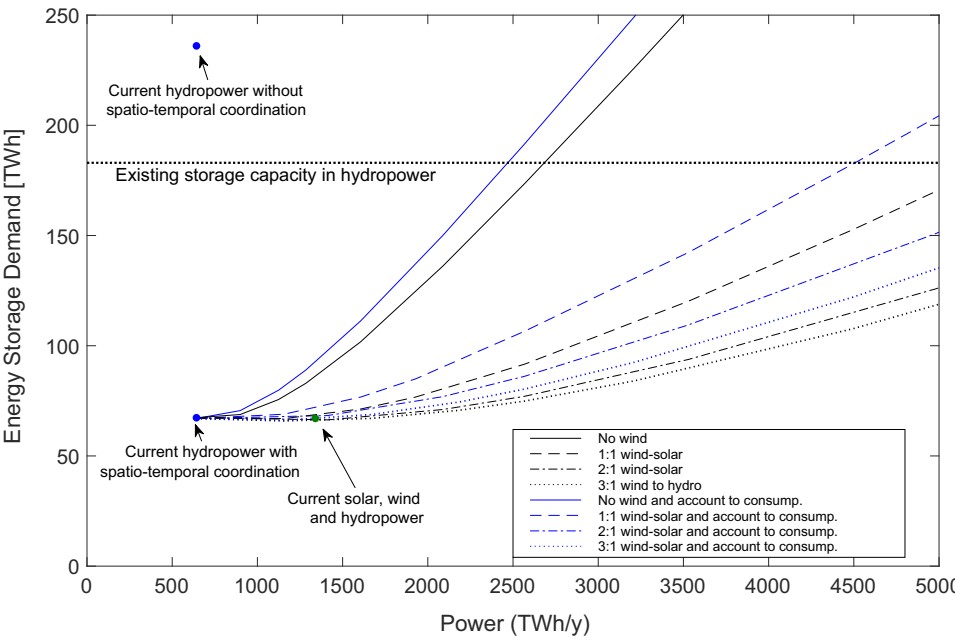

**Fig. 3 Scenarios for energy storage demand.** Energy storage demand associated with various expansion scenarios of the solar-wind-hydropower production system utilizing a constant hydropower system with current production capacity and increasing production of solar and wind power in different proportions. These scenarios consider a fully aggregated or coordinated production system with no restrictions in the power transmission grid.

energy storage gain for PV solar, wind and hydropower over Europe"). The analyzed scenarios assume that the installed power capacity is the same as the long-term consumption, i.e., no wasting of potential electricity production occurs. However, renewable energy production is often characterized by a certain degree of curtailment by which the actual power capacity is generally higher than the actual utilized power. In practice, the higher production capacity in comparison to the required energy consumption can reduce the energy storage demand even further.

As mentioned in the introduction, the current operational management models do not explicitly consider incentives for virtual energy storage gain, which is generally expressed as a suboptimization of a delimited electric production mark[19]. However, we argue that not only can a fully renewable electric production system be designed and engineered, but it also inherits management services in the form of spatiotemporal coordination and energy source complementarity, which increases in value with the size of the management domain. Implementation of these services requires coordinated seasonal forecasts of the continental energy components and economic market rules that resolve curtailments resulting from suboptimizations of operational management.

## Discussion and conclusions

North Atlantic Oscillation causes periodic climate behavior over the European continent, with alternating cold-dry and mild-wet conditions in Northern Europe and the opposite alternation in the south. This climatic behavior leads to long-term variations in weather patterns, which in combination with short-term intermittent variations in renewable energy availability, can lead to energy droughts with limited production capacity in solar-wind-hydropower systems. However, this study shows that within Europe, virtual energy storage gain (VESG) that is many times larger than that available in current hydropower reservoirs can be realized through spatiotemporal coordination of renewable energy production. The strong complementarity between these renewable energy sources should be utilized alongside spatio-temporal coordination efforts that increase the gain continuously

with the aggregation scale and the regulatory period considered. Solar, wind and hydropower can at all times meet the entire electricity consumption of the European region, 4950 TWh y$^{-1}$, without spilling potential energy production by using existing energy storage available in hydropower in combination with both spatiotemporal coordination and appropriate resource complementarity. We find that suitable shares of solar, wind and hydropower that maximize VESG is about 2:4:1, leading to potential storage gains of 298 TWh from spatiotemporal management and 169 TWh from complementarity (section "Virtual energy storage gain for PV solar, wind and hydropower over Europe" and Supplementary Note 4). It is worth noting that the physical energy storage demand was found to be 131 TWh, which stresses the important role of the existing hydropower for the reliability of renewable energy production. The spectral decomposition of the energy storage shows that the energy storage demand increases with the regulatory period considered all though the magnitude in the power deficit decreases. For example, the energy storage demand is about doubled as the regulatory period is tripled from 5 to 15 years. The probability that the energy storage demand exceeds the available hydropower storage is a cumulative effect of power deficits governed by the spectrum of climate periods, which is easier to foresee from observations than sudden extreme events. Thus, on time horizons beyond 3–5 years this study indicates a possible need for introducing reserve power successively with time. These findings are based on assumptions of no energy spillage, no electric transmission limitations and a regulatory time horizon limited to 5 years.

The fact that virtual energy storage grows proportionally to the amounts of renewables can facilitate the transition towards a renewable electrical power production system. While VESG can be achieved to a limited extent on a national energy market, the major effects arise at the continental scale at distances up to approximately 1200 km. Our study also highlights the strong complementarity effects in an energy mix of hydropower, solar power and wind power across Europe, which is reflected in a negative or very low covariance on the yearly period between the potentials for solar and wind power, as well as solar power and

electric consumption. The cross- and autocovariance spectra depend on the distribution of power plants, which can have implications for future scenarios of renewable energy expansion over the entire continent. Here, we present several scenarios for the expansion of solar-wind-hydropower production systems with different shares of solar and wind power while keeping the current hydropower production capacity constant. For Europe, we found that 2–3 times more wind power than solar power provides an appropriate complement to existing hydropower systems, resulting in a minimum energy storage demand. The energy storage demand assessed here arises at a management horizon of about 5 years, whereas accounting for decadal-long modes of variability in the climate system could require additional reserve power capacity or management schemes covering even larger areas than considered here (Supplementary Note 3).

Knowledge of the covariances, and hence the complementarity, between solar power, wind power and hydropower is important for designing suitable energy mixes and technical capacities for energy storage, as well as for electric transmission. However, there are remaining challenges to incorporating this understanding in both energy system design and operational management. Existing planning and dispatch models applied to the continental scale are generally based on optimization theory that takes into consideration technical solutions for energy storage, such as hydrogen storage in overground steel tanks, bioenergy and nuclear powe[12,13]. While optimization of continental energy systems acknowledges a huge optimization space both in terms of spatial representation of power plants and resolution of the time horizon, the spectral approach used here addresses the maximum potential (or bound) for VESG within a coordination distance. To fully utilize the economic potential of spatiotemporal management and the complementary characteristics of solar, wind and hydropower, there is a need for suitable incentives and geographically extended, common market rules for virtual services, as well as continental hydroclimatic forecasts that account for covariances between renewable potential. Utilization of the VESG arises predominantly at the continental scale and relies on stable international energy markets and very long, common electrical transmission lines across the European continent. This finding introduces several additional important questions regarding the risks related to malfunctioning and international political stability. More homogenization or interactions between the energy markets can support the development of a market for VESG. Such homogenization would simultaneously address issues of energy availability and affordability and hence be related to some of the concerns that have been expressed about "energy justice"[37].

## Methods: technical scenarios, analytical approach and data support
**Demand-production scenarios**. The electricity production from solar, wind and hydropower within EU27/28 was approximately one-third of the total electricity consumption in 2020[38]. For 48 countries in Europe and parts of the Middle East, hydropower production accounted for approximately 640–650 TWh $y^{-1}$ in 2020, and its share of electricity generation was fairly constant[39]. Here, we consider hydropower as a constant resource of 642 TWh $y^{-1}$ at the current power plant locations that were identified in 362 of the 1055 subareas used in this study (section "Hydropower potential at existing station locations"). Figure 1a shows the spatial distribution of power production for the scenario with equal proportions of solar, wind and hydropower, which covers 40% of the total demand. Existing hydropower was complemented by an increasing production share from solar and wind power up to the entire current electricity consumption of 4950 TWh $y^{-1}$ in the study domain. Solar and wind power

represented 195 and 503 TWh $y^{-1}$ in 2020, respectively, but can be significantly expanded in the future to meet the current EU energy policy and its roadmap for sustainable economies called the "European Green Deal". The technical scenarios for power production potential are based on 35–40 years of historic records of daily hydroclimatic information obtained from the Copernicus ECMWF database for an area of $13.6 \times 10^6$ km$^2$, which covers most parts of Europe and the Middle East, as well as technical data for the existing hydropower and wind power systems of Europe and solar panel efficiencies.

A uniform distribution of a PV solar power potential was assumed in 995 of the 1055 subareas (section "Solar power potential"). For instance, PV solar power of the same amount as hydropower (642 TWh $y^{-1}$) corresponds to a uniform area coverage fraction of PV cells of $3.77 \times 10^{-4}$ (–) or 5127 km$^2$ of (horizontal) land covered by PV panels. There are no strict material or areal limitations for PV panel development, so here, we extend the fraction up to 4 times more (or 2568 TWh $y^{-1}$), requiring approximately 10% of Europe's urban area of approximately 200,000 square kilometers[40]. The corresponding scenario for wind power was defined by recognizing the estimated potential power time series at more than 15,000 onshore wind farm locations distributed in 477 of the 1055 European subareas and by scaling these time series linearly (section "Wind power potential at existing station locations"). Such scaling would correspond to the development of the number of power plants without considering actual spatial limitations for such development.

In Europe, the current growth rates of solar and wind power are approximately 20 and 30 TWh $y^{-1}$, respectively, and therefore, a long period is expected to fulfill electricity production needs with only solar, wind and (constant) hydropower. Nevertheless, there is a rapidly growing degree of expansion of these resources worldwide; hence, there is a need to assess the appropriate combination and location of renewable electricity production. Consequently, the production scenarios are based on 1834 power units (995 PV, 477 wind, 362 HP) with a fluctuating power production potential that at all times should match the consumption within an electrical transmission-distribution range (without curtailment). The consumption scenario considered here replicates the historical monthly consumption pattern of the 19 EU countries between 2009 and 2022 (Supplementary Note 1). This approach omits the short-term intermittency of demand-production matching[41], but reflects typical seasonal variations governing the predominant energy storage demand. For comparison, we also consider a constant electricity consumption, as it would facilitate a first measure of the complementarity of renewable energy production and the demand for energy storage.

**Hydropower potential at existing station locations**. The average total hydropower potential considered in the hydrological model domain is 365.7 TWh $y^{-1}$ at the current GranD station locations, accounting for all dams and reservoirs with a storage capacity of more than 0.1 cubic kilometers[42]. This system includes 1377 hydropower stations within the pan-European E-HYPE hydrological model domain[43] used to estimate historical water runoff data (Supplementary Note 1). Furthermore, the corresponding power time series were scaled to match the observed power production at all hydropower stations without direct connections to large reservoirs, which is common in cascading hydropower, and finally set to a constant level of 642 TWh $y^{-1}$ distributed on 362 of 1055 areas shown as pixels in Fig. 1. Although there is potential to increase the production by up to 8–9% at existing hydropower stations using different strategic measures[44], this study assumes constant production from hydropower. The daily

runoff for a 35-year period (1981–2015) is simulated for 35,408 watersheds covering Europe and parts of the Middle East. Model outputs have shown robustness for climate change impact assessments[45,46], seasonal hydrological predictions[47,48] and energy applications[33].

**Solar power potential**. Hydroclimatic data were obtained from the Copernicus ECMWF database for an area of $13 \times 10^6$ km², which covers most parts of Europe and the Middle East (Fig. 1). The total area was the same as that used for the simulation of historical runoff time series and the estimation of hydropower potential. Daily data of the downwards surface solar radiation (SSRD) from January 1, 1979, to December 31, 2020, were converted to radiation incident on a fixed, south-facing panel with an inclination equal to the latitude and further covering the PV power potential. The power potential was averaged over 24 h (both night and day) considering the changes in solar elevation and azimuth angles and aggregated for 995 of a total of 1055 uniformly distributed subareas (Supplementary Note 1). The SSRD is the equivalent horizontal shortwave radiation and was used as incident power density input in the PV production power formula by Huld et al.[49,50]. These estimates of the power potential were performed for cadmium-telluride PV cells and for a hypothetical uniform area coverage (see scenario description below); however, the general behavior of crystalline silicon PV cells would be similar in the context here. The conversion efficiency depends on both temperature and wind velocity, which was accounted for by using the hydroclimatic data from the Copernicus database.

**Wind power potential at existing station locations**. Meteorological data relevant to wind power potential were obtained from ERA5, which is a reanalysis product of the ECMWF's General Circulation Model available in the Copernicus Climate Data Store. For comparison, data were also taken from MERRA 2 and JRA 55 and used to derive wind speed time series from January 1, 1979, to December 31, 2019, at the locations of more than 15,000 onshore wind farms from the "World Wind Farm Database". At the locations of existing wind farms (Supplementary Note 1) and technical specifications of power plants, we estimated the current annual power potential (351.7 TWh y⁻¹). This estimation recognized the existing heights and extent of wind power plants, as well as the near-ground atmospheric boundary layer and surrounding topography. The physical variables were downscaled and interpolated (using the inverse distance weighting and nearest neighbor methods) to plant locations, and eventually, the daily averaged power output was calculated using the power curve of the "General Electric 2.5 MW turbine". Finally, the results were spatially aggregated on the same rectangular regions used for solar power potential (left-hand side panel of Fig. 1), whereby only 477 of the 995 regions (or pixels) contained wind power.

**Spectral decomposition of the power potential and the energy storage demand**. Time series of renewable energy potentials aggregated over Europe show both high short-term intermittency and seasonal variations. Although this is interesting, it does not define the cumulative amount of energy deficits determining the energy storage demand, which results from both the magnitude of power deviations and their durations. Spectral decomposition can facilitate the analysis of the energy storage demand as a function of the spectra of covariances existing spatiotemporally (Supplementary Note 2). In general, the energy storage demand can be expressed as an integration of the deviations between the power production and demand over the spectrum of frequencies. This relationship can be expressed by using an energy balance spectral decomposition[33] (Supplementary Note 2):

$$Std(E) = \sqrt{2 \int_{f_1}^{f_2} \frac{S(P_\delta)}{(2\pi f)^2} df} \qquad (1)$$

where $Std$ denotes the standard deviation, $E$ is the instantaneous energy storage (J), $f$ is the frequency of variations (s⁻¹), and $S(P_\delta)$ is the power spectral density of $P_\delta$. Here, $P_\delta$ is the power deviation between potential production and consumption (W), i.e., $P_\delta = P_{min} - P_C$, with $P_{min}$ being the minimum power with zero need for dispatching/wasting energy potential (W), and $P_c$ being the power consumption. Since hydroclimatic time series are generally both periodic and apparently random, Eq. (1) opens for the possibility of excluding periodic variations longer than $1/f_1$ while accounting for the remaining variance. This management strategy could be motivated if the production scheme is complemented by additional long-term energy reserve capacities. Following the arguments of Supplementary Note 3 one can show that the maximum energy storage demand $E_{max} = C\ Std(E)$, where $C$ is a coefficient in the range of 1–1.5 corresponding to the exceedance return period and the regulatory period of 3–5 years. Hence, the standard deviation of the instantaneous energy storage will subsequently be referred to the energy storage demand, $E_{SD}$ (J).

The minimum power is defined here as a theoretical value since electricity production from renewables often underestimates what is potentially possible due to the installed turbine capacity, spillage due to dispatch constraints (curtailment) and lack of energy storage[20]. While spectral decomposition greatly simplifies the statistical assessment of the regulatory energy demand, a challenging task is to assess the power spectral density of the power deviations considering the coordination scheme applied to the spatial distribution of production units and the characterizing covariances of the continental climate system. For this purpose, we use the historical hydroclimatic time series to define the demand-production scenarios for the 1834 power units (section "Demand-production scenarios"). Such data reflect the observed statistical behavior of the climate system, including cross- and autocovariances of the different renewable energy components.

Furthermore, we adopt a management strategy that applies spatiotemporal production-demand that matches within a coordination distance, $R$ (m or decimal degrees), without considering transmission constraints or other curtailment reasons. Production for power unit pairs farther than $R$ is considered to be fully statistically independent (see section A2.2 in SI for details). Hence, the computational procedure involves assessing the covariance spectral densities of the power potential and demand for 3.36 million pairs ($1834 \times 1834$) of power units. Thus, the analysis is focused on scenarios of solar, wind and hydropower production potential that precisely match the electricity consumption without any global dispatch constraints. Therefore, the energy storage satisfies the following conservation equation: $\partial E/\partial t = P_{min} - P_c$, which is expressed in spectral form according to Eq. (1).

## Data availability
Data used and produced in this paper and data reports are available at the following repository at Zenodo.org, https://doi.org/10.5281/zenodo.7750145. Historical data obtained through E-HYPE are openly available on the HYPEweb portal (https://hypeweb.smhi.se/explore-water/historical-data/europe-time-series/).

## Code availability
All codes used in this study to assemble and mathematically treat the available data are available on request as Matlab scripts.

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

## Acknowledgements

This study was funded by the project "The role of hydropower as a regulating resource in a renewable energy system with climate impact and increased internationalization of electricity markets" granted by the Swedish Energy Agency (Grant no. 52095-1). I.P. was also supported by the EU Horizon 2020 project E-SHAPE (EuroGEOSS Showcases: Applications Powered by Europe) under Grant Agreement 820852. A.W. and J.R. were also supported by StandUP for Energy strategic research framework. C.B.U. is also supported by the Academy of Finland under the HYDRO-RI project and the Freshwater Competence Centre – FWCC.

## Author contributions

A.W.: initiator, conceptualization, theory development and analyses, writing original manuscript, data documentation. DM: data aggregation and manuscript review, data documentation. JR: conceptualization, manuscript review and writing. IP: conceptualization, manuscript review and writing. CBU: conceptualization, manuscript review and writing.

## Funding

## Competing interests

The authors declare no competing interests.
