## [Peer Review File · Communications Engineering]

Reviewers' comments:

Reviewer #1 (Remarks to the Author):

This paper investigates the extent to which the connection and coordination of wind, solar, and hydropower over large spatial scales in continental Europe can effectively reduce the capacity of energy storage needed to balance the continental electric load demand, represented by a Virtual Energy Storage Gain.

In general, I do not have any issue with the methods or analysis itself, but rather that the points made by this study do not seem to be new. The literature on energy systems shows numerous examples of how enabling wind, solar, and other resources to coordinate over large spatial scales reduces the variability in net electric load for the combined region, resulting in reduction in energy storage capacities needed to achieve increasing targets for greenhouse gas or zero-carbon electricity penetration. Some of the examples in the literature even quantify this using grid modeling, which is a step more detailed than the profile analysis presented in this paper. A relevant example of previous work for the U.S. is presented by Shaner et al (<https://doi.org/10.1039/C7EE03029K>), and multiple examples on modeling coordinated grids in Europe by Tom Brown (TU Berlin). Therefore, the main issue here is that I struggle to understand what is novel about this analysis since it doesn't specify how this analysis method or other aspects of the study contribute understanding about energy systems planning that is not already established in the literature.

Therefore, I recommend that the authors make it more clear how this study and its methods provide additional understanding regarding the notion that large-scale spatiotemporal coordination among renewable resources can reduce energy storage needs. Is it the scale or the granularity for example? This is a prerequisite for being considered for publication.

As a minor note: in the manuscript file I reviewed, it seems Figure 4 is missing even though Figure 4a/b/c is referenced in the text.

Reviewer #2 (Remarks to the Author):

- The paper performs a spectral analysis of long-term production in hydro power, solar power, wind power and power demand.

It analyzes the complementarity of energy carriers and tries to quantify the amount of storage that could be saved by using complementary production units. This "virtual storage" is found to be significantly higher than the actual physical storage.

- Moreover, the exposition of technical details in the appendix seems unfinished. I was not satisfied with the derivations, they appear to have been rushed under time pressure and need more attention to the details to become readable.

- Figure S3 is not readable due to the different curves overlapping. Conventionally this type of figure, is represented with all generations stacked, while consumption is overlaid. This allows to easily read-off whether the sum of generation covers the demand. Please adjust the figure accordingly.

- "Turkey" instead of "Türkiye"

- Appendix A2.1 needs to be reworked significantly.

The section appears unfinished and full of contradicting passages.

Consequently, it is extremely hard to discern the meaning of the derivation.

I can not entirely possible to judge whether it is correct or not due to odd naming conventions.

This may be a language problems, however several used terms do not match the meaning in the equation.

For instance, the passage opens with "For an energy system without export or import," and later introduces "P_{dis} = dispatched energy due to spillage and export", where export is introduced. This contradicts the introductory sentence.

Equation (1) is derived from (A1) - (A3). However, the definition used for (A1) are off.

"P_{dis} = dispatched energy due to spillage and export" is a bit of a misnomer, as it is not related to dispatch. It collects the terms for "export" and "spillage" or "curtailment". Dispatch arises when power plants are dispatched to satisfy demand and is probably reflected better by P_c. I advise to write this explicitly and as two separate terms -P_{export} - P_{spillage} or something similar instead of P_{dis}.

"P = installed potential for power production (W)" is not consistent with the use of P in (A1).

As (A1) describes dE/dT ie the change in energy storage, P needs to be the actual production not the installed capacity (which i believe installed potential means). If "export+curtailment" and "consumption" exceeds actual production P, storage shrinks and vice versa. Hence the meaning of (A1) is retained and P can only be actual Production.

- When reading the technical appendix, i get the impression that the authors make numerous assumptions when deriving equation (1).

For instance, "E_{max}" seems not to be a true maximum but an approximation where a single standard deviation is allowed.

This approach is valid, but loses resilience to extreme events.

In particular, a single standard deviation, is not a very high level of security. Depending on the underlying distribution assumptions.

For normal distributions, a standard deviation lies within 68% of all events, which seems a rather low margin of error for power systems.

From what i gather the authors assume Gaussianity, which is a strong assumption. Please correct me if i understood wrongly.

It could well be that the author derive all results empirically, but i was not able to catch that exactly

from the appendix.

Wind power on single sited is usually described by Weibull distributions with significant covariances, thus assuming that aggregate production is Gaussian is a stretch, as we do not have easy I.I.D. guarantees.

The authors need to make their distribution assumptions and all other technical assumption such as safety of a single standard deviation clear in the main text.

- While the papers methods seem solid, the fact that long-distance transmission allows to save storage is well known within the energy community. As such im not convinced of the papers innovativeness. In particular, i feel that the appendices need to be reworked, before submitting to any journal as there are numerous inconsistencies. It may also be advisable to move some of the appendices in the main text. I find the particular effort to quantify "virtual storage" interesting from a technical point of view. However, i feel that it may be more relevant to a specialized journal such as Applied Energy or Energy.

Point-by-point response to referee's comments

Reviewer 1

- Comment: In general, I do not have any issue with the methods or analysis itself, but rather that the points made by this study do not seem to be new Some of the examples in the literature even quantify this using grid modeling, which is a step more detailed than the profile analysis presented in this paper. A relevant example of previous work for the U.S. is presented by Shaner et al., and multiple examples on modeling coordinated grids in Europe by Tom Brown (TU Berlin). ... Therefore, I recommend that the authors make it more clear how this study and its methods provide additional understanding regarding the notion that large-scale spatiotemporal coordination among renewable resources can reduce energy storage needs. Is it the scale or the granularity for example?

Response: As highlighted by the reviewer, two main aspects of the novelty of this study include the importance of the spatial aggregation scale and the spectral decomposition of the regulatory time horizon for the virtual energy storage gain. Figure panel 1 demonstrates how the energy storage demand decreases continuously with coordination distance (the spatial scale). Furthermore, it is shown in our study that solar, wind and hydropower production can meet the entire electricity consumption of Europe, 4,950 TWh/y over a regulatory horizon of 3 - 5 years, by using only existing energy storage available in hydropower in combination with spatio-temporal coordination and appropriate resource complementarity from wind and solar power. The quantification of the storage gain, the estimation of the optimal mix of renewable resources and the appropriate regulatory horizon are generally important information for motivating the continental management as well as for guiding the technological development.

To account for the reviewer's comment, we have stressed more clearly in the Abstract, Introduction and Findings that the spectral approach facilitates the quantification of virtual energy storage gain as well as separates the problem on the characteristic periods used in the energy system management.

Moreover, we incorporate two new references based on the reviewer's suggestions; these are discussed in the third paragraph of the introduction. While these previous studies and other studies, that are already referenced in the paper have found that continental management of renewable energy sources are beneficial to reduce the resource variability, here we quantify this gain in terms of the virtual energy storage and show how it increases continuously with the coordination / aggregation scale. For example, the study by Shaner et al. (2018) analyses the covariation in solar and wind power as well as the implications for the reliability of meeting production targets. However, our study provides a quantification of the bounding energy storage demand as function of aggregation scale and time horizon of the regulation. Spectral decomposition of amplitude and duration in the resources variability facilitates the quantification of the virtual energy storage as function of the aggregation scale and temporal scale.

- Comment: Figure 4 is missing even though Figure 4a/b/c is referenced in the text.

Response: The text should have read "Figure 3" and we have corrected this.

Reviewer 2

- Comment: Moreover, the exposition of technical details in the appendix seems unfinished. I was not satisfied with the derivations, they appear to have been rushed under time pressure and need more attention to the details to become readable.

Appendix A2.1 needs to be reworked significantly. The section appears unfinished and full of contradicting passages. ... several used terms do not match the meaning in the Equation.

Response: We generally agree with the reviewer and have significantly improved the entire Supplemental Information significantly, including the nomenclature in section A2.1, and, in response to the questions raised below, the statistics of energy storage demand (Appendix A3). The Supplementary Information has also undergone a language review by the American Journal Experts.

- Comment: "P = installed potential for power production (W)" is not consistent with the use of P in (A1). As (A1) describes dE/dt ie the change in energy storage, P needs to be the actual production not the installed capacity (which i believe installed potential means).

Response: We agree with the reviewer and have modified section A2.1 accordingly. We use the following terminology: P = actual power production (W), P_s = spilled power (not produced while available) and otherwise exported energy (not considered as consumption), P_c = power consumption (W).

- Comment: Figure S3 is not readable due to the different curves overlapping. Conventionally this type of figure, is represented with all generations stacked, while consumption is overlaid.

Response: OK! We have produced a new Figure S3 with solar, wind and hydropower potential stacked in the diagram and overlay the demand curve. Due to the large magnitudes in short-term variations in these resources it was necessary for graphical clarity to use a monthly moving average of these time series as is clarified in the figure caption. All analyses are, however, conducted using daily values.

- "Turkey" instead of "Türkiye"

This could be a journal policy and the current policy could possibly differ between language regions. However, it has been decided by the United Nations to use the

name “Türkiye”. Wikipedia has published a discussion of this topic, including references to relevant UN documents and a statement made by the US State Department: https://en.wikipedia.org/wiki/Name_of_Turkey#cite_note-4

- Comment: When reading the technical appendix, I get the impression that the authors make numerous assumptions when deriving equation (1). For instance, "E_max" seems not to be a true maximum but an approximation where a single standard deviation is allowed. This approach is valid, but loses resilience to extreme events.

Wind power on single sited is usually described by Weibull distributions with significant covariances, thus assuming that aggregate production is Gaussian is a stretch, as we do not have easy I.I.D. guarantees.

Response: We have included an explanation of the relationship between the maximum energy storage demand and the standard deviation in the energy storage demand in the first paragraph of section 2.2. This section also references Appendix 3 in the Supplemental Information, which we have extended to provide more thorough explanation of the statistical interpretation of the energy storage demand.

Since, hydro-climatic time-series are generally combinations of periodic and apparently random components, we separate these two contributions to the variance. Further, it is shown that the maximum energy storage demand E_{max} is close to equal to $Std(E)$ at a regulatory period of 3 – 5 years.

In Appendix 3 we discuss the event probabilities of energy drought periods, which are related to the resilience to extreme events. We have tested both Gaussian and Weibull probability density functions to describe the distribution of power and energy storage as well as in the estimation of exceedance probabilities for the storage. Results of both distributions are now presented in Appendix 3.3. Both distribution functions show similar goodness of fit to both power and storage.

- Comment: The authors need to make their distribution assumptions and all other technical assumption such as safety of a single standard deviation clear in the main text.

Response: While the assumptions are mentioned where introduced, we summarized the most essential ones at the end of the first paragraph of the Findings section: *“These findings are based on assumptions of no energy spillage, no electric transmission limitations and a regulatory time horizon limited to 5 years”*.

- Comment: While the papers methods seem solid, the fact that long-distance transmission allows to save storage is well known within the energy community.... I find the particular effort to quantify "virtual storage" interesting from a technical point of view.

Response: The main, novel contribution of our paper refers to the quantification of virtual energy storage as function of aggregation scale and the temporal period considered in the regulation. The study also shows that the current hydropower offers sufficient energy storage for a fully renewable electricity system if spatio-temporal coordination and complementarity are accounted for. These findings should provide quantitative motivation for a continental management of the renewable electrical production system and guidance for realizing virtual energy storage by spatio-temporal coordination and introduction of appropriate shares of renewables. See also the above responses to the first point raised by Reviewer 1.

REVIEWERS' COMMENTS:

Reviewer #1 (Remarks to the Author):

The authors have done a good job clarifying the novelty of the paper relative to the original version, which was my main comment.

I do understand that this study was conducted under somewhat idealized conditions from a power system representation standpoint to focus on the concept of virtual energy storage gain. The paper can benefit from some discussion of how some of the constraints considered in more detailed power system models may affect the results.

Overall, however, the revised manuscript is much improved and is suitable for publication.

Reviewer #2 (Remarks to the Author):

I see that the authors worked to improve the appendix and the manuscript.

I also see some improvements in the exposition of the maths, but for instance the derivation and use of (A4) by means of $F(i,j)$ still seems a bit opaque. What exactly is the functional form of F ? Is (A4) equivalent to the equation in the flow text?

Nonetheless, I reemphasize that it is in principle a well executed and technical sound manuscript.

However, I note that both I and the other referee had concerns about the novelty of the paper, as it reports on common knowledge in the field.

The authors reply: "Response: The main, novel contribution of our paper refers to the quantification of virtual energy storage as function of aggregation scale and the temporal period considered in the regulation."

I agree that the particular quantification strategy is interesting. However, it represents a technical variation on well known facts. As such it is not well suited in a venue of broad interest to all engineering, but would fit a specialized journal targeting energy research in general and a community interested in the technicalities of energy system research in particular.

I reiterate my recommendation to submit to Energy, Applied Energy or possibly a suitable IEEE journal.

The study also shows that the current hydropower offers sufficient energy storage for a fully renewable electricity system if spatiotemporal coordination and complementarity are accounted for."

This conclusion comes with taking several very strong assumptions such as no spillage and free unbounded energy transfer. I work at a TSO (National Grid Provider) and am in principle too of the opinion, that additionally energy transport capacity will curb storage demand. This is as I said well

known. However, expanding grid infrastructure seems currently similarly challenging as expanding production infrastructure.

Mostly due to lengthy legal processes required for building new power lines, but of course also cost considerations need to be considered. The perspective of the grid expansion problematic is disregarded in this conclusion, which is a simplification that can be made but it is a strong assumption.

Consequently, the optimistic results of the paper are somewhat diminished by the strong assumptions needed for their validity.

Overall, I stand by my initial verdict: reject due to lack of novelty and broad interest, and recommend resubmission in a specialized journal due to technical sound work that merits publishing.

2nd rebuttal letter for “Continental complementarity of renewable energy mixes”.

We thank the two reviewers and the Editors for providing valuable feed-back on the manuscript "Continental Complementarity of Renewable Energy Mixes", which has significantly improved it. In the last round of reviews the second reviewer was asking about a clarification of the functional form of F used in Eqn. (A4) in the Supplemental Information. This function takes the value 1 when the cross-correlation between two power stations is considered or the value 0 when the two power stations are operating independently. This can be expressed by means of the Heaviside function, which is now introduced for clarity.

We note that reviewer 2 feel that the paper advances over existing knowledge and could be published in specialized journal. To some extent we concur, but would like to stress that the main findings of this study are novel and has a general relevance to the adaptation of the society to a sustainable future; Solar, wind and hydropower alone can, in principle, meet the entire current electricity demand of Europe based solely on energy storage in existing hydropower and without wasting energy through curtailment in high-energetic weather conditions. We show through spectral analysis that spatio-temporal coordination and complementarity of the renewable resources across Europe can induce a virtual energy storage gain that is several times larger than that available in existing hydropower reservoirs.